# Obesity and Morbidity Risk in the U.S. Veteran

**DOI:** 10.3390/healthcare8030191

**Published:** 2020-06-29

**Authors:** Jose A. Betancourt, Paula Stigler Granados, Gerardo J. Pacheco, Ramalingam Shanmugam, C. Scott Kruse, Lawrence V. Fulton

**Affiliations:** School of Health Administration, Texas State University, San Marcos, TX 78666, USA; psgranados@txstate.edu (P.S.G.); gjp46@txstate.edu (G.J.P.); shanmugam@txstate.edu (R.S.); scottkruse@txstate.edu (C.S.K.); lf25@txstate.edu (L.V.F.)

**Keywords:** U.S. veteran health, obesity, comorbidities, risk-factors, overweight, diabetes

## Abstract

The obesity epidemic in the United States has been well documented and serves as the basis for a number of health interventions across the nation. However, those who have served in the U.S. military (Veteran population) suffer from obesity in higher numbers and have an overall disproportionate poorer health status when compared to the health of the older non-Veteran population in the U.S. which may further compound their overall health risk. This study examined both the commonalities and the differences in obesity rates and the associated co-morbidities among the U.S. Veteran population, utilizing data from the 2018 Behavioral Risk Factor Surveillance System (BRFSS). These data are considered by the Centers for Disease Control and Prevention (CDC) to be the nation’s best source for health-related survey data, and the 2018 version includes 437,467 observations. Study findings show not only a significantly higher risk of obesity in the U.S. Veteran population, but also a significantly higher level (higher odds ratio) of the associated co-morbidities when compared to non-Veterans, including coronary heart disease (CHD) or angina (odds ratio (OR) = 2.63); stroke (OR = 1.86); skin cancer (OR = 2.18); other cancers (OR = 1.73); chronic obstructive pulmonary disease (COPD) (OR = 1.52), emphysema, or chronic bronchitis; arthritis (OR = 1.52), rheumatoid arthritis, gout, lupus, or fibromyalgia; depressive disorders (OR = 0.84), and diabetes (OR = 1.61) at the 0.95 confidence interval level.

## 1. Introduction

The average American today is overweight or obese: a trend that represents a threat to individuals and to American society as a whole [1,2,3]. Across the demographics of the general U.S. population, this rise in average weight for individuals may be attributable to a variety of reasons, including sedentary lifestyles, poor eating habits, lack of exercise and individuals simply not making good health choices [4,5]. According to a study that reviewed data from 1980 to 2008, the number of obese people doubled over a two decade period from 1.3 million to 2.6 million globally, primarily due to unhealthy habits (e.g., consumption of tobacco and alcoholic beverages), unhealthy diet (e.g., energy drinks, excess salt and sugar, intake of high saturated fat, and discretionary foods), and physical inactivity [5]. Obesity and its co-morbidities are a major cause of morbidity and mortality in the United States [3]. It is linked to a variety of other poor health outcomes including cardiovascular disease (CVD), hypertension (HTN), type 2 diabetes mellitus (T2DM), hyperlipidemia, stroke, certain cancers, sleep apnea, liver and gall bladder disease, osteoarthritis, and gynecological problems [6,7,8,9,10]. Each of these problems results in poor health outcomes for the affected individual in the long term: outcomes that are costly to the individual, costly to their families, and ultimately costly to the U.S. healthcare system [10]. Furthermore, as this unwelcome trend impacts all ethnic, socio-economic, and geographic regions of the U.S. population, one particular sub-group which displays higher levels of obesity is made up of those who served in the U.S. Armed Forces: the U.S. Veteran [11,12,13,14]. Particularly striking is the fact that not only is there a disproportionate number of Veterans who are obese, but there also exists an overall disproportionate poorer health status among elderly Veterans when compared to the health of the older non-Veteran population enrolled in Medicare-managed care programs [15]. The Veterans Health Administration (VHA) has attempted to tackle obesity through a national, population-based approach since early 2000, utilizing a number of different methods, some more effective than others [6,15]. Of the six million patients who receive VHA care yearly, 80% fall into the categories of either being overweight or obese [16]. Additionally, a recent study found that over 44% of Operation Enduring Freedom, Operation Iraqi Freedom, or Operation New Dawn (OEF/OIF/OND) Veterans are obese (body mass index (BMI) > 30 kg/m^2^), which exceeds the national obesity prevalence rate of 39% in people younger than age 45 years of age [17]. Unfortunately, this trend is not dissipating as exhibited in another study that found that obesity rates for Veterans doubled from 14% to 32% between 2001 and 2008 [18]. For a group of individuals whose high level of physical fitness was a paramount requirement of employment while on active duty, there may be unique factors to this population causing them to become obese later in life. Title 38 of the U.S. Code of Federal Regulations defines a Veteran as “a person who served in the active military, naval, or air service and who was discharged or released under conditions other than dishonorable” [19]. This definition explains that any individual who completed a term of military service for any branch of the U.S. Armed Forces (e.g., Army, Navy, Marines, Air Force) is classified a “Veteran” as long as they received an honorable discharge. According to a 2019 estimate conducted by the National Center for Veterans Analysis and Statistics, there are an estimated 18.3 million living U.S. Veterans [20]. Another study performed in 2018 provided an updated estimate of the prevalence of obesity in U.S. military Veterans (estimated 32.7%), and evaluated a broad range of sociodemographic, military service, physical and mental health, and lifestyle characteristics associated with obesity in this population [21]. The results of this study paint a sobering picture of this growing trend among the U.S. Veteran population, however, it also highlights certain sub-groups within this population such as black women Veterans, women Veterans with schizophrenia, younger Veterans (under age 65), and Native Hawaiian/Other Pacific Islander and American Indian/ Alaska Native Veterans—who are more at risk, as well as exhibit certain behaviors which correlate to poor health indicators [11]. 

Of particular concern, the Veteran population may suffer not only from obesity but additional risk factors such as post-traumatic stress disorder (PTSD) [9] and depression [22,23], which may further compound the overall health risk of the individual [24]. The prevalence of obesity among Veterans with PTSD is higher than the prevalence of obesity among Veterans overall within the VHA (47% vs. 41%, respectively) [25]. Studies have indicated that PTSD is also associated with an increased risk of T2DM, a known risk factor for obesity [9]. Overall, the health outcomes of the Veteran population in the U.S. lag behind their non-Veteran counterpart, perhaps as a result of the hardships associated with military service [26]. 

The individual military service experience for each Veteran varies by such factors as military occupation, time in service, number and types of deployments, and assorted stressors such as whether the Veteran served in a combat environment and/or experienced family separation. One unique stressor that all Veterans experience at the end of their military service (no matter how long or how short of a period they served) is their transition from a regimented environment and subsequently adjusting to a new way of life once their military service comes to an end [20]. These cumulative stressors may impact the Veteran in a variety of ways. In fact, more than 43% of Veterans surveyed in a recent study had a mental health diagnoses, with depressive disorders, PTSD, and substance abuse disorders being the most prevalent [15]. Military exposures, such as multiple deployments and exposure to combat, contribute to challenges in the re-integration to civilian life for all Veterans. Factors that contribute to the increased risk of obesity include changes in eating patterns/eating disorders, changes in physical activity, physical disability, and psychological comorbidity [22,27,28]. 

The research presented here seeks to improve the understanding of the commonalities as well as the differences in obesity rates and associated co-morbidities between the U.S. Veteran population and the nation as a whole. We investigated whether the average Veteran rate of obesity/overweight is similar to that of the U.S. non-Veteran population in the United States when controlling for demographics (age, race/ethnicity, gender, and marital status) and socioeconomics (income, education, and employment), and hypothesized that Veterans are likely to be more obese/overweight. Secondarily, we looked at the comorbidities associated with obesity, speculating that Veterans are likely to have higher rates of coronary heart disease (CHD) or angina; stroke; skin cancer; other cancers; chronic obstructive pulmonary disease (COPD), emphysema, or chronic bronchitis; arthritis, rheumatoid arthritis, gout, lupus, or fibromyalgia; depressive disorders, kidney disease (not including stones or bladder infections); and diabetes. Recently published data, from the U.S. Centers for Disease Control and Prevention (CDC) Behavioral Risk Factor Surveillance System (BRFSS), provide a rich trove of information that may aid the public health community as well as the VHA in both identifying at-risk sub-groups within the Veteran population, as well as developing health interventions aimed at addressing these threats to their health. 

## 2. Materials and Methods 

### 2.1. Data

For this study, we leveraged the 2018 BRFSS. These data are considered by the CDC to be the nation’s best source for health-related survey data, and the 2018 version includes 437,467 observations. Applying complex weighting to these observations results in a picture of the entire nation’s self-reported health [29,30]. 

### 2.2. Independent Variables 

Research Question 1 (RQ1): obesity/overweight status of Veteran vs. non-Veteran

Four variables served as controls for demographics including age in 6-year groups, race/ethnicity based on CDC coding, gender, and marital status. The variables and their associated questions and factor levels are found in Table 1. The age variable included CDC imputations for missing data (less than 2%) as did the race/ethnicity question. For gender, 0.16% of the population responded “Don’t Know/Not Sure” or “Refused” (0.14% when weighted). These observations were imputed with the modal response, “Female.” Gender was then recoded as a dichotomous variable per Table 1. Only 0.74% (0.76% weighted) of the observations for marital status were “Refused” or BLANK, and these were imputed with the mode (“Married”).

Socio-economic status included measures for income level, education level, and employment status. Table 2 provides those variables and factor levels. The proportion of “Don’t Know/Not Sure”, “Refused”, and BLANK was 16.56% for income (16.8% weighted). Because of the large proportion in these categories, they were left for a separate evaluation as an individual factor level. For education, only 0.36% were in the “Refused” or BLANK categories (0.39% weighted), and these values were imputed with the median as well. The employment status question had 0.86% (1.01% weighted) “Refused” or BLANK. The mode response of “Employed for wages” was imputed.

The independent variable of most interest was Veteran status (Table 3). This categorical variable included a single question. Factor levels 7 (Don’t Know) and 9 (Refused) were 0.2% of the population (0.19% when weighted). These values were then imputed with the modal response, “No.” The recoding of this variable is reflected in Table 3.

### 2.3. Dependent Variables for RQs 1 through 10

RQ1 assesses the obesity/overweight status as a function of the independent variables. The dependent variable of interest for RQ1 is the obesity/overweight status, and this is measured as a categorical variable in the BRFSS variables of interest for RQ as shown in Table 4. Factor levels 3 (Overweight) and 4 (Obese) are combined into a single factor (level = 1), and the factor levels 1 (Underweight) and 2 (Normal Weight) are the complement (level = 0) for the dependent variable. The approximately 8% of missing observations are imputed with the modal response (“Overweight”). The dependent variable is then a dichotomous measure. For the remaining 9 dependent variables (RQ 2–10), all were also dichotomously coded. Missing responses were small in all cases and coded to the modal response of “No.”

### 2.4. Model and Methods

For all RQ’s, we modeled the dependent variable as a function of demographics, socio-economic status, and Veteran status. Because of the categorical nature of the variables and the fact that survey weights are applied which allows for integer variables to have fractional values, quasi-binomial distribution was used. The quasi-binomial distribution allows for the fractional values of integer variables and is thus an improvement over a binomial [31,32,33,34]. Since complex survey data weights often result in non-integer values, the quasi-binomial (depicted in Equation (1)) is the appropriate link function. In this equation, *p* is the probability of overweight/obese status (weighted), *N* is the number of weighted observations, *k* is the (likely non-integer) number of successes, and *ϕ* is the variance that is not accounted for by the binomial distribution. Analyses were performed using the *survey* package for complex weighting [34] in R Statistical Software [35]:(1)$$P\left(X=k\right)=\left(\begin{array}{c} N\\ k \end{array}\right)p{\left(p+k\phi \right)}^{k-1}{\left(1-p-k\phi \right)}^{n-k}$$

## 3. Results

### 3.1. Descriptive Statistics

#### 3.1.1. Dichotomous Variables 

Proportions for the weighted dichotomous variables are shown in Table 5 along with differences for the Veteran and non-Veteran population and relative risk (RR) ratios (Veteran to non-Veteran). (Standard errors are all less than 0.001.) Overall, the population is comprised of 47.4% male and 12.1% Veteran. About 69.7% of the population report obesity/overweight status, 5.2% coronary heart disease (CHD), 4% stroke, 8.2% skin cancer, 8.4% other cancers, 7.7% chronic obstructive pulmonary disease (COPD), 29.9% arthritis, 18.4% depression, 3.6% kidney disease, and 12.9% diabetes. When comparing the proportions, Veterans have higher rates of BMI (75.8% vs. 68.8%, RR = 1.102, odds ratio (OR) = 1.420), arthritis (38.0% vs. 28.7%, RR = 1.324, OR = 1.523), diabetes (18.3% vs. 12.2%, RR = 1.500, OR = 1.612), skin cancer (14.7% vs. 7.3%, RR = 2.014, OR = 2.188), other cancers (12.8% vs. 7.8%, RR = 1.641, OR = 1.735), CHD (10.8% vs. 4.4%, RR = 2.455, OR = 2.631), kidney disease (4.9% vs. 3.4%, RR = 1.441, OR = 1.464), and stroke (6.5% vs. 3.6%, RR = 1.806, OR = 1.862). Surprisingly, a lower proportion of Veterans reported depression (16.3% vs. 18.7%, RR = 0.872). These proportions do not tell the entirety of the story, as the Veteran population contains more than double the male vs. the female population (89.3% vs. 41.6%), and thus the actual differences must account for gender as well as other demographics. Figure 1 shows the distribution of dichotomous dependent variable proportions segmented by Veteran status. 

#### 3.1.2. Other Non-Geographical Factor Variables

Table 6 shows the proportions for factor variables with greater than two levels along with relative risk. Overall, the modal responses indicate that the population was largely 65 and older (28.5%), white (67.8%), married (52.8%), earning $75K+ (30.9%), a college graduate (40.2%), and employed for wages (46.1%). Veterans were much more likely than non-Veterans to be 65 and older (RR = 1.954) and much less likely to be Asian (RR = 0.370) or Hispanic (RR = 0.441). Veterans were also more likely to be divorced (RR = 1.232) and much less likely to earn less than $10K (RR = 0.430). Veterans were also much more likely to be retired (RR = 2.115).

### 3.2. Inferential Statistics

For RQ1, the quasi-binomial model produced the results shown in Table 6. The largest effect size is associated with the age between 45 to 55 years old (OR = 3.23). Hispanics were much more likely to be overweight or obese (OR = 1.724, *p* < 0.001) along with males (OR = 1.503, *p* < 0.001). After accounting for demographic and socioeconomic variables and Veteran status odds ratio (1.12, *p* < 0.001); those who refused to answer the income question, were not sure, or left it blank, were more likely to be obese (OR = 1.238, *p* < 0.001) along with those that reported less than a ninth grade education (OR = 1.319, *p* < 0.001). North Dakotans were more likely to self-report overweight or obese status (1.222).

For RQs 1 through 10, we present the odds ratios for Veteran status along with their associated 95% confidence levels. For all of these research questions, there was an odds ratio greater than one after evaluating the socio-economic, demographic, and geographic variables; however, kidney disease was not statistically different from the non-Veteran population. Table 7 provides these results.

For arthritis, Veteran status resulted in an odds ratio of 1.26 (Veteran to non-Veteran). For depression, however, the odds ratio is 1.26. The directionality reversed by accounting for covariates, and the effect size (OR = 1.29), is appreciable. Diabetes, skin cancer, and other cancer had odds ratios of 1.10, 1.18, and 1.38 (respectively), while CHD, COPD, and stroke were 1.32 1.46, and 1.28, respectively. In all cases, Veterans were at higher odds for these morbidities. 

## 4. Discussion 

These findings show not only a significantly higher risk of obesity in the U.S. Veteran population, but also a significantly higher level of the associated co-morbidities when compared to non-Veterans. Other studies have looked at obesity and co-morbidities among Veterans, showing a higher prevalence of co-morbidities among the obese members of this population [21]. However, our study illustrated that when comparing obesity and other health outcomes to non-Veterans, Veterans are more likely to be overweight or obese (OR = 1.12) and to suffer from a negative health outcome as a result. Like many large health systems trying to address the obesity epidemic, the VHA healthcare system offers a number of weight-management programs in an effort to stem the tide of this obesity epidemic [6]. A number of VA initiatives aimed at providing the tools for the Veteran to better manage their weight include education on proper nutrition, the benefit of regular exercise, the use of technology such as daily apps, and when warranted, bariatric surgery. However, even with this increased emphasis by the VA, these data show that some of these initiatives are having mixed results in terms of actual weight reduction, compliance by the Veteran and ultimately in meeting the needs of the targeted Veteran. 

The research clearly shows that obesity is associated with a dearth of illnesses, many of which we looked at in this study [34,35]. In fact, individuals who are obese are more likely to have deleterious effects on pulmonary function, respiratory mechanics, gas exchange, control of breathing, and exercise capacity which result in respiratory conditions [36,37,38] such as obstructive sleep apneas (OSAS), asthma, chronic obstructive pulmonary disease (COPD), emphysema and chronic bronchitis [13,39,40,41,42]. Our study showed that Veterans are much more likely to suffer from a pulmonary disease (OR = 1.47) when compared to non-Veterans. Exposure to burn-pits, smoking or other factors that may impact the respiratory system might be a factor in this higher likelihood of experiencing pulmonary issues [43,44]. Obesity has also been associated with the risk of developing certain cancers [45,46] including colon-rectal cancer, esophageal adenocarcinoma, thyroid cancer and renal cancers [40,47,48]. It is important to note that in our study, Veterans were more likely to have skin cancers (OR = 1.18) and other types of cancers (OR = 1.38) when compared to non-Veterans. The higher odds of Veterans having cancer may be related to exposures or other service-related activities [49]. Other chronic diseases associated with obesity include osteoarthritis (OA) and osteoporosis (OP) [50,51] are mainly due to their systemic effects on the body, owing to metabolic and inflammatory alterations [52]. Veterans in our study were shown to be more likely to suffer from conditions associated with arthritis (OR = 1.26) [53]. Obesity is also associated with depression where the relationship between these conditions is often bidirectional: the presence of one increases the risk of developing the other [54,55,56]. Interestingly, our study showed that a lower proportion of Veterans reported depression (16.3% vs. 18.7%, RR = 0.872). However, as it was noted, the Veteran population surveyed contains more men than women (89.3% vs. 41.6%), and thus the actual differences must account for gender and how each report depression [55,56,57]. Multiple studies have shown that individuals who are overweight or obese were significantly associated with a higher risk of moderately/severely impaired kidney function/kidney disease [58,59,60]. In our study, kidney disease was not significantly higher than the non-Veteran population. Obesity is closely associated with individuals who suffer from diabetes [61]. Our study also showed that Veterans are more likely to have diabetes (OR = 1.10) and although this is heavily associated with obesity, there is a possibility that it could be a result of occupational exposures [62,63,64].

### Limitations

One limitation in this study is that the data originate from the BRFSS which is comprised of self-reported answers to questions which cannot be independently verified. Additionally, self-reported data contain several potential sources of bias that should be noted as limitations including selective memory (only remembering certain aspects of answer), telescoping (inaccurately recalling events that occurred at a different time), attribution (recalling positive attributed to self but attributing negative events to external entities), and exaggeration (the embellishment of events).

## 5. Conclusions

The findings outlined in this study accentuate the problem that obesity has on the health of the Veteran population in the U.S.: a problem that impacts more Veterans than the average American and is continuing to grow in magnitude. However, our study findings also pointedly highlight a number of additional health conditions that, when taken in concert with obesity, paint a concerning picture of the overall health risks or morbidities suffered by the U.S. Veteran population. These morbidities among Veterans could prompt the VA to re-evaluate the current listing of certain disabilities that the VA presumes were caused by military service (presumptive Service connection). This is because of the unique circumstances of a specific Veteran’s military service and the potential for certain Service-unique exposures. If a presumed condition is diagnosed in a Veteran in a certain group, they can be awarded disability compensation. Future studies should examine the interaction of multiple comorbidities on health outcomes of this population, specifically the interaction of depression, diabetes and obesity. Additionally, it would be valuable to further refine and pinpoint geographic locations where Veteran populations in the U.S. congregate, so as to better resource health intervention programs where they can provide the most benefit to target populations. These results can better inform the VHA and other Veteran-serving institutions on the magnitude of the overall health needs of the Veteran population and help develop more effective health interventions aimed at improving the overall health of the Veteran. By analyzing population-based health data for Veterans when compared to the U.S. population, this might better identify the population of U.S. military Veterans (by specific demographic characteristics) who may benefit from health system interventions aimed at solutions for reducing weight and reducing morbidity from certain health risks [12]. These study results may also provide information aimed at raising the awareness of communities which may have a significant portion of their community comprised of Veterans. By raising awareness, community interventions may be formulated in order to provide weight management tools and health interventions aimed at assisting these deserving Americans whose years of sacrifice may still be suffering from invisible wounds resulting from military service. 

## Figures and Tables

**Figure 1 healthcare-08-00191-f001:**
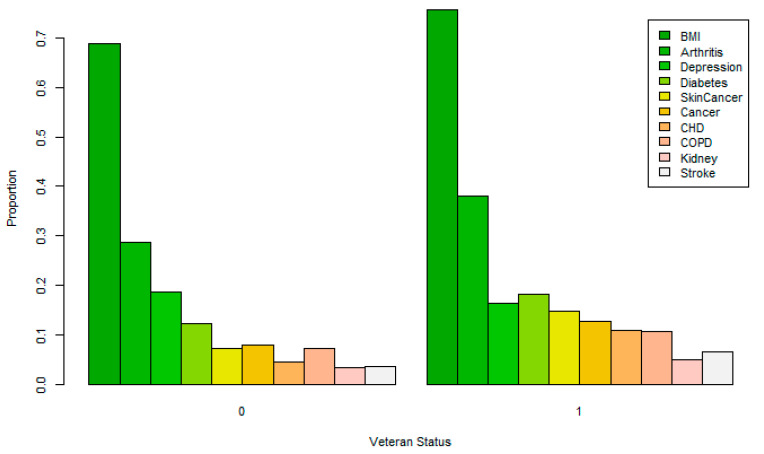
Proportion of obesity and the associated comorbidities by Veteran status.

**Table 1 healthcare-08-00191-t001:** Behavioral Risk Factor Surveillance System (BRFSS) Coding variables for age, race/ethnicity, gender and marital status.

BRFSS Variable Name	Question	Code
AGE_G	Six-level imputed age category	1—18 to 24 2—25 to 34 3—35 to 44 4—45 to 54 5–55 to 64 6—65+
IMPRACE	Imputed race/ethnicity value	1—White 2—Black 3—Asian 4—American Indian/Alaskan 5—Hispanic 6—Other non-Hispanic
SEX1	What is your sex? or What was your sex at birth? Was it…	0—Female 1—Male
MARITAL	Are you (marital status)?	1—Married 2—Divorced 3—Widowed 4—Separated 5—Never married 6—A member of an unmarried coupled

**Table 2 healthcare-08-00191-t002:** Socio-economic variables for Research Question 1 (RQ1).

BRFSS Variable Name	Question	Code
INCOME2	Is your annual household income from all sources?	1—<$10K 2—$10K ≤ Income < $15K 3—$15K ≤ Income < $20K 4—$20K ≤ Income < $25K 5—$25K ≤ Income < $35K 6—$35K ≤ Income < $50K 7—$50K ≤ Income < $75K 8—$75K or more 9—Don’t Know/Not Sure /Refused/BLANK
EDUCA	What is the highest grade or year of school you completed?	1—None or Only Kindergarten 2—Grades 1 through 8 3—Grades 9 through 11 4—Grades 12 or GED 5—College 1 to 3 years 6—College 4+ years (Graduate)
EMPLOY1	Are you currently…?	1—Employed for Wages 2—Self-Employed 3—Out of Work ≥1 Year 4—Out of Work <1 Year 5—Homemaker 6—Student 7—Retired 8—Unable to Work

**Table 3 healthcare-08-00191-t003:** Veteran status variable for RQ1.

BRFSS Variable Name	Question	Coded Levels
VETERAN3	Have you ever served on active duty in the United States Armed Forces, either in the regular military or in a National Guard or military reserve unit?	0—No 1—Yes

**Table 4 healthcare-08-00191-t004:** Comorbidity variables and levels.

Variable	Question from Codebook	Recoded Response
BMI5CAT	Four categories of body mass index (BMI)	0—No 1—Yes
CVDCRHD4 (RQ2)	(Ever told) you had angina or coronary heart disease?	0—No 1—Yes
CVDSTRK3 (RQ3)	(Ever told) you had a stroke.	0—No 1—Yes
CHCSCNCR (RQ4)	(Ever told) you had skin cancer?	0—No 1—Yes
CHCOCNCR (RQ5)	(Ever told) you had any other types of cancer?	0—No 1—Yes
CHCCOPD1 (RQ6)	(Ever told) you have chronic obstructive pulmonary disease, emphysema or chronic bronchitis?	0—No 1—Yes
HAVARTH3 (RQ7)	(Ever told) you have some form of arthritis, rheumatoid arthritis, gout, lupus, or fibromyalgia? (arthritis diagnoses include: rheumatism, polymyalgia rheumatica; osteoarthritis (not osteoporosis *(sic)*); tendonitis, bursitis, bunion, tennis elbow; carpal tunnel syndrome, tarsal tunnel syndrome; joint infection, etc.)	0—No 1—Yes
ADDEPEV2 (RQ8)	(Ever told) you have a depressive disorder (including depression, major depression, dysthymia, or minor depression)?	0—No 1—Yes
CHCKDNY1 (RQ9)	(Ever told) you have kidney disease? (Do NOT include kidney stones, bladder infection or incontinence.)	0—No 1—Yes
DIABETE3 (RQ10)	(Ever told) you have diabetes (If ´Yes´ and respondent is female, ask ´Was this only when you were pregnant? (Pregnancy-Related coded as 0)	0—No 1—Yes

**Table 5 healthcare-08-00191-t005:** The relative risk and odds ratio for comorbidities.

	87.9%*	12.1%*		
Variable	Not Veteran	Veteran	Relative Risk	Odds Ratio
BMI	0.688	0.758	1.102	1.420
Arthritis	0.287	0.380	1.324	1.523
Depression	0.187	0.163	0.872	0.847
Diabetes	0.122	0.183	1.500	1.612
Skin Cancer	0.073	0.147	2.014	2.188
Cancer	0.078	0.128	1.641	1.735
Coronary Heart Disease (CHD)	0.044	0.108	2.455	2.631
Chronic Obstructive Pulmonary Disease (COPD)	0.073	0.107	1.466	1.522
Kidney	0.034	0.049	1.441	1.464
Stroke	0.036	0.065	1.806	1.862

* Percentage of total population.

**Table 6 healthcare-08-00191-t006:** Socioeconomic differences between the Veterans and non-Veterans with the relative risk and odds ratio.

Variable	Estimate	SE	*t* Value	Pr (>|t|)	Odds Ratio
(Intercept)	−0.090	0.192	−0.47	0.640	
25 to 34	0.613	0.030	20.34	<0.001	1.846
35 to 44	0.971	0.032	30.43	<0.001	2.641
45 to 54	1.115	0.032	34.53	<0.001	3.051
55 to 64	1.066	0.032	33.11	<0.001	2.903
65+	0.954	0.035	27.54	<0.001	2.595
Black	0.495	0.024	20.39	<0.001	1.641
Asian	−0.425	0.039	−11.04	<0.001	0.654
American Indian/Alaskan	0.265	0.055	4.78	<0.001	1.303
Hispanic	0.544	0.026	21.05	<0.001	1.724
Other Non-Hispanic	0.100	0.035	2.84	0.005	1.105
Male	0.408	0.014	29.61	<0.001	1.503
Divorced	−0.144	0.020	−7.28	<0.001	0.866
Widowed	−0.241	0.023	−10.41	<0.001	0.786
Separated	−0.129	0.044	−2.94	0.003	0.879
Never married	−0.256	0.020	−12.91	<0.001	0.774
A member of an unmarried coupled	−0.104	0.034	−3.08	0.002	0.901
$10K ≤ Income < $15K	0.143	0.043	3.3	<0.001	1.154
$15K ≤ Income < $20K	0.132	0.040	3.26	0.001	1.141
$20K ≤ Income < $25K	0.157	0.040	3.95	<0.001	1.170
$25K ≤ Income < $35K	0.122	0.039	3.12	0.002	1.129
$35K ≤ Income < $50K	0.172	0.038	4.49	<0.001	1.188
$50K ≤ Income < $75K	0.160	0.038	4.17	<0.001	1.173
8-$75K or more	−0.015	0.037	−0.41	0.685	0.985
9-Don’t Know/Not Sure/Refused/BLANK	0.214	0.036	5.91	<0.001	1.238
Grades 1 through 8	0.277	0.187	1.48	0.138	1.319
Grades 9 through 11	0.005	0.184	0.03	0.979	1.005
Grades 12 or GED *	0.047	0.183	0.26	0.796	1.049
College 1 to 3 years	0.073	0.183	0.4	0.692	1.075
College 4+ years (Graduate)	−0.228	0.183	−1.24	0.215	0.796
Self-Employed	−0.250	0.023	−11.09	<0.001	0.778
Out of Work ≥1 Year	−0.055	0.046	−1.21	0.227	0.946
Out of Work <1 Year	−0.059	0.043	−1.37	0.170	0.943
Homemaker	−0.269	0.030	−9.09	<0.001	0.764
Student	−0.330	0.037	−8.92	<0.001	0.719
Retired	−0.170	0.021	−7.95	<0.001	0.844
Unable to Work	−0.001	0.028	−0.02	0.982	0.999
Alaska	−0.204	0.064	−3.19	0.001	0.815
Arizona	−0.217	0.047	−4.62	<0.001	0.805
Arkansas	0.036	0.051	0.71	0.480	1.037
California	−0.263	0.040	−6.52	<0.001	0.769
Colorado	−0.410	0.041	−10.01	<0.001	0.664
Connecticut	−0.164	0.041	−4.03	<0.001	0.849
Delaware	0.005	0.049	0.11	0.913	1.005
District of Columbia	−0.483	0.048	−9.96	<0.001	0.617
Florida	−0.151	0.045	−3.36	0.001	0.860
Georgia	−0.063	0.041	−1.53	0.126	0.939
Guam	−0.125	0.076	−1.64	0.100	0.882
Hawaii	−0.266	0.046	−5.78	<0.001	0.766
Idaho	−0.147	0.055	−2.69	0.007	0.863
Illinois	−0.075	0.046	−1.62	0.104	0.928
Indiana	−0.075	0.044	−1.7	0.089	0.928
Iowa	0.089	0.041	2.14	0.032	1.093
Kansas	0.074	0.041	1.82	0.069	1.077
Kentucky	0.042	0.049	0.87	0.387	1.043
Louisiana	0.023	0.049	0.46	0.647	1.023
Maine	−0.108	0.044	−2.45	0.014	0.897
Maryland	−0.072	0.039	−1.85	0.065	0.931
Massachusetts	−0.196	0.044	−4.47	<0.001	0.822
Michigan	−0.026	0.041	−0.63	0.530	0.975
Minnesota	−0.042	0.038	−1.11	0.266	0.959
Mississippi	0.151	0.049	3.08	0.002	1.162
Missouri	−0.081	0.049	−1.67	0.095	0.922
Montana	−0.188	0.050	−3.76	0.000	0.828
Nebraska	0.068	0.042	1.64	0.102	1.071
Nevada	−0.128	0.060	−2.12	0.034	0.880
New Hampshire	−0.128	0.047	−2.72	0.007	0.880
New Jersey	−0.145	0.064	−2.25	0.024	0.865
New Mexico	−0.273	0.047	−5.81	<0.001	0.761
New York	−0.187	0.038	−4.95	<0.001	0.830
North Carolina	−0.067	0.049	−1.36	0.173	0.935
North Dakota	0.200	0.049	4.07	<0.001	1.222
Ohio	0.030	0.042	0.71	0.480	1.030
Oklahoma	0.032	0.048	0.66	0.512	1.032
Oregon	−0.171	0.046	−3.76	0.000	0.843
Pennsylvania	−0.050	0.046	−1.08	0.280	0.952
Puerto Rico	−0.408	0.053	−7.73	<0.001	0.665
Rhode Island	−0.121	0.048	−2.54	0.011	0.886
South Carolina	0.005	0.042	0.11	0.910	1.005
South Dakota	0.079	0.056	1.4	0.162	1.082
Tennessee	−0.030	0.051	−0.59	0.558	0.970
Texas	−0.053	0.050	−1.06	0.291	0.949
Utah	−0.178	0.040	−4.39	<0.001	0.837
Vermont	−0.295	0.046	−6.43	<0.001	0.744
Virginia	−0.086	0.042	−2.04	0.041	0.918
Washington	−0.114	0.039	−2.88	0.004	0.893
West Virginia	0.155	0.049	3.16	0.002	1.168
Wisconsin	0.018	0.051	0.36	0.722	1.018
Wyoming	−0.195	0.050	−3.88	<0.001	0.823
Veteran?	0.112	0.021	5.24	<0.001	1.119

* GED: General Educational Development test for high-school equivalence.

**Table 7 healthcare-08-00191-t007:** Odds ratios and confidence intervals by RQ.

Research Question	Comorbidity	Lower CI	Odds Ratio	Upper CI
RQ1	BMI	1.07	1.12	1.17
RQ2	Arthritis	1.21	1.26	1.31
RQ3	Depression	1.24	1.29	1.35
RQ4	Diabetes	1.05	1.10	1.16
RQ5	Skin Cancer	1.12	1.18	1.25
RQ6	Cancer	1.30	1.38	1.47
RQ7	CHD	1.24	1.32	1.41
RQ8	COPD	1.37	1.46	1.56
RQ9	Kidney Disease	0.99	1.08	1.19
RQ10	Stroke	1.17	1.28	1.39

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
