# Peer review of "Obesity and Morbidity Risk in the U.S. Veteran"

_healthcare, 2020, doi:10.3390/healthcare8030191_

Round 1
Reviewer 1 Report
The present study by Betancourt J et al examined the prevalence of obesity and morbidity risk between veteran and non-veteran population. This is collected and analyzed based on 2018 Behavioral Risk Factor Surveillance System (BRFSS). From this data set author conclude that veteran population are significantly higher risk for obesity, in addition, they are also higher risk for comorbid conditions such as cancer, stroke, diabetes, arthritis, etc. It is well written study with insights. I do not have any comments for this article.
Author Response
Thank you so much for the time it took for you to review our article. We are glad you liked it.
Reviewer 2 Report
I am not sure why RR was used.
The abstract can be improved with OR and 95%.
I highlighted it in yellow color.

Author Response
Reviewer comment 1. I am not sure why RR was used.
Author response: Thank you for pointing this out and we apologize for not being more clear on why we selected to use Relative Risk (RR). Relative risk produces and is calculated directly from the ratio or proportions in the tables. A good discussion of the difference is found at the following website: https://www.ncbi.nlm.nih.gov/pmc/articles/PMC4640017/. Conversion to Odds Ratio (OR) is indeed possible, however this will likely overestimate the risk. Please let me know if we are still unclear on why we chose to utilize RR.
Reviewer comment 2. The abstract can be improved with OR and 95%. I highlighted it in yellow color.
Author response: Thank you for your valuable recommendation. We have inserted the odds ratios (OR) after each of the appropriate variables. Additionally, we have inserted the OR in Tables 5 and 6 as well as in the appropriate text locations in addition to relative risk. Finally, we have made the recommended changes highlighted in yellow. We are confident that these additional modifications will improve not only the abstract but the manuscript as well. Thank you again.
Reviewer 3 Report
This is an interesting research question about the prevalence of obesity and comorbidities in veterans versus non veterans in the US. However, the way the data is presented is not very correct and needs to be changed.
Major comments:
- Table 1: The way variables are presented is not very correct. The categories and the way variables were analysed is not a research question. The title of the table is to compare obesity and overweight status in veterans versus non veterans which is not really shown.
- Tables 1 and 2 could be combined and variables of interest compared in veterans versus non veterans.
- The coding of the BRFSS variables does not need to be presented in the article, it is hard to follow and not always clear.
- Table 5, please show what is being presented in gender and cancer and spell CHD and COPD in the table’s legend. What are the percentages 87.9% and 12.1% are they % of veterans and non-veterans of the total population? If so, this should be clearly stated in the table.
- In the introduction, the authors discuss the difference in types of service, number of deployments etc. in veterans per se. It would be more interesting to analyse separately groups of veterans according to these differences, and stratify for the duration of service in order to explain the higher prevalence of obesity and overweight in this particular group. Would it be related to a higher time service or not?
Minor comments:
- Abstract line 19, typo: data instead of date.
- Correct table 1 and line 119 (age levels are 6 and not 5).
Author Response
Reviewer comment 1. This is an interesting research question about the prevalence of obesity and comorbidities in veterans versus non veterans in the US. However, the way the data is presented is not very correct and needs to be changed. Table 1: The way variables are presented is not very correct. The categories and the way variables were analysed is not a research question. The title of the table is to compare obesity and overweight status in veterans versus non veterans which is not really shown. Tables 1 and 2 could be combined and variables of interest compared in veterans versus non veterans.
Author response: Thank you for your comment and for pointing this out. Our intent in Tables 1 and 2 was to identify the independent variables vs. dependent variables separately. The method in which we present the data is for transparency in the variables we used and the ability to replicate the findings for those interested in doing so. For better ease of reading, we have moved these tables to the Appendix and we have changed the title of Table 1. And finally, in terms of comparing Veteran to non-Veteran, we have provided that in the descriptive statistics. We are confident that this presents the variables in a more understandable manner.
Reviewer comment 3. The coding of the BRFSS variables does not need to be presented in the article, it is hard to follow and not always clear.
Author response: Thank you for your recommendation. We have moved Tables 1 and 2, which present the coding of the BRFSS variables into the appendix section of the manuscript.
Reviewer comment 4. Table 5, please show what is being presented in gender and cancer and spell CHD and COPD in the table’s legend. What are the percentages 87.9% and 12.1% are they % of veterans and non-veterans of the total population? If so, this should be clearly stated in the table.
Author response: Thank you for your recommendations. We have removed the variable ‘gender’ from this table as the relevance did not fully contribute to the main points of this table. We have fully spelled out ‘Coronary Heart Disease (CHD)’ and ‘Chronic Obstructive Pulmonary Disease (COPD)’ on lines 173-175 in the main body of the manuscript immediately preceding Table 5 in order to define what the acronyms mean in the table. We also placed an asterix after the numbers ’87.9%’ and ’12.1%’ as well as added text on line 187 to better inform the reader that these numbers refer to percentage of total population.
Reviewer comment 5. In the introduction, the authors discuss the difference in types of service, number of deployments etc. in veterans per se. It would be more interesting to analyse separately groups of veterans according to these differences, and stratify for the duration of service in order to explain the higher prevalence of obesity and overweight in this particular group. Would it be related to a higher time service or not?
Author response: Thank you for these very insightful recommendations. While we agree with the Reviewer that by separating and stratifying the Veteran groups by the recommended variables, unfortunately the BRFSS database lacks this level of granularity necessary for this analysis to be conducted. The BRFSS simply asks whether the survey respondent is a Veteran or not and does not go further than that. We are hopeful that future studies may utilize other data sources containing these levels of detail so that we may answer such questions as whether time in service makes a difference in obesity and overweight levels.
Reviewer comment 6. Abstract line 19, typo: data instead of date, and correct table 1 and line 119 (age levels are 6 and not 5).
Author response: Thank you for pointing these errors out. We have made the recommended changes.
Round 2
Reviewer 3 Report
The authors have taken my comments into consideration, I have no further suggestions.